

# Not so sluggish: the success of the *Felimare picta* complex (Gastropoda, Nudibranchia) crossing Atlantic biogeographic barriers

Frederico Almada, André Levy and Joana I. Robalo

MARE–Marine and Environmental Sciences Centre, ISPA Instituto Universitário de Ciências Psicológicas, Sociais e da Vida, Lisbon, Portugal

## ABSTRACT

The molecular phylogeny of the Atlanto-Mediterranean species of the genus Felimare, particularly those attributed to the species *F. picta*, was inferred using two mitochondrial markers (16S and COI). A recent revision of the Chromodorididae clarified the taxonomic relationships at the family level redefining the genus *Felimare*. However, conflicting taxonomic classifications have been proposed for a restrict group of taxa with overlapping morphological characteristics and geographical distributions designated here as the *Felimare picta* complex. Three major groups were identified: one Mediterranean and amphi-Atlantic group; a western Atlantic group and a tropical eastern Atlantic group. *F. picta* forms a paraphyletic group since some subspecies are more closely related with taxa traditionaly classified as independent species (e.g. *F. zebra*) than with other subspecies with allopatric distributions (e.g. *F. picta picta* and *F. picta tema*). Usually, nudibranchs have adhesive demersal eggs, short planktonic larval phases and low mobility as adults unless rafting on floating materials occurs. Surprisingly however, the phylogeny of the *F. picta* complex suggests that they successfully cross main Atlantic biogeographic barriers including the mid-Atlantic barrier. This ability to cross different biogeographic barriers may be related to *F. picta*'s distinct life history and ecological traits. Compared to other Chromodorididae *F. picta* has larger eggs and planktotrophic larvae which could be related to a longer planktonic phase.

## INTRODUCTION

Nudibranchs are heterobranch mollusks (Gastropoda) that comprise more than 3000 species (*Willan & Coleman, 1984*). The family Chromodorididae includes some of the most striking colored nudibranchs present in almost all marine habitats. Conspicuous aposematic colorations advertise defensive adaptations that include the production or incorporation of bioactive chemicals from cnidarians or sponges upon which they feed (*Wollscheid & Wägele, 1999*; *Haber et al., 2010*). These toxins have captured the interest of numerous authors (*Gaspar, Rodrigues & Calado, 2009*; *Haber et al., 2010*; *Cruz, Gaspar & Calado, 2012*) and represent a source of natural compounds whose biological activity is being actively prospected. As a consequence, chromodorids are also good models for the

Corresponding author
Frederico Almada,
frederico.almada@gmail.com

study of color pattern evolution and mimicry in marine species (*Gosliner & Johnson, 1999*). Clarifying the evolutionary relationships among chromodorids would pave the way to predict characteristics that are still undescribed and would help to disentangle genetic from environmental effects (e.g. *Gaspar, Rodrigues & Calado, 2009*).

The taxonomy of the chromodorid nudibranchs was originally based on morphological data (*Rudman, 1984*). *Gosliner & Johnson (1999)* revised the phylogeny of the worldwide genus *Hypselodoris*, including some of the species that are more abundant in the northeastern Atlantic and Mediterranean Sea. These revisions were followed by a number of morphological (*Alejandrino & Valdés, 2006*) and molecular studies (*Wollscheid & Wägele, 1999*; *Wollscheid-Lengeling et al., 2001*; *Turner & Wilson, 2007*; *Johnson, 2011*), including the description of several new species (*Dacosta, Padula & Schrödl, 2010*; *Ortigosa & Valdés, 2012*). *Johnson & Gosliner (2012)*, based on two mitochondrial genes, proposed a new phylogenetic hypothesis of the chromodorid and reorganized its traditional taxonomy. These authors identified the Pacific Ocean as an effective biogeographic barrier, proposing the inclusion of all eastern Pacific, Atlantic and Mediterranean *Hypselodoris*, together with two species from the genus *Mexichromis* from the eastern Pacific and Caribbean, into a different genus, *Felimare*, while retaining the west/central Pacific *Hypselodoris* species within this genus. Eastern Pacific, Atlantic and Mediterranean species previously in the genera *Chromodoris* and *Glossodoris* were included within the *Felimida*. Prior to their revision, *Hypselodoris*, *Chromodoris* and *Glossodoris* were the most species rich chromodorid genera. Recently, using additional species of these genera and a nuclear DNA marker, *Ortigosa et al. (2014)* recovered a polytomy between several Atlantic and Mediterranean *Felimida* and *Felimare* species.

## Taxonomic considerations of *Felimare picta*

Within the Chromodorididae *F. picta* presents the wider distribution area throughout the tropical and subtropical Atlantic Ocean, including the Mediterranean Sea and the eastern Atlantic archipelagos of Cape Verde, Canaries, Madeira and Azores (*Ortea, Valdés & García-Gómez, 1996*). These coastal sea slugs are one of the largest chromodorid species and vary greatly in colour and pattern, resulting in the traditional classification under many species names (e.g. *Felimare* (*Hypselodoris*) *webbi* (d'Orbigny, 1839) and *Felimare* (*Hypselodoris*) *tema* (*Edmunds, 1981*)). Variation in external morphology or internal anatomy also led to incongruencies in the taxonomic classification of *F. picta* subspecies (*Ortea, Valdés & García-Gómez, 1996*; *Alejandrino & Valdés, 2006*). One of the cases that illustrates the ambiguous situation of this group is the changing taxonomic status of the southwest Atlantic representatives classified as subspecies (*Felimare picta lajensis*) by *Troncoso, Garcia & Urgorri (1998)*, as a valid species (*Felimare lajensis*) by *Domínguez, García & Troncoso (2006)* and then downgraded again to a subspecific level by *Dacosta, Padula & Schrödl (2010)*. Similarly, *Felimare tema*, originally described from Ghana by *Edmunds (1981)* was reclassified as subspecies (*Felimare picta tema*) by *Ortea, Valdés & Garcia-Gomez (1996)*. Color pattern similarities between *Felimare picta verdensis* and *F. picta tema* (sensu *Ortea, Valdés & Garcia-Gomez, 1996*) also raised some doubts on the validity of these subspecies. Table S1 in Supplementary Data summarizes the main

taxonomic classifications and also the overlapping distribution areas of some of these subspecies.

The phylogenetic relationships of the *F. picta* complex with other taxa long recognised as distinct species, both from the east and west Atlantic, remains largely unknown. *F. picta* is the only amphi-Atlantic species of this genus, but along the east Atlantic shores and central Atlantic islands we can find several species with overlapping distributions. Along the west Atlantic shore this phylogenetic relationship is even more complex, with a close trans-isthmian phylogenetic relationship between all eastern Pacific and the Atlantic/Mediterranean *Felimare* species, together with two species previously assigned to the genus *Mexichromis* (*F. porterae* and *F. kempfi*) reported by *Johnson & Gosliner (2012)*.

## Biogeography

The ability to cross biogeographic barriers is normally restricted to highly mobile species or species that produce propagules with high potential to disperse at least during a particular phase of their life-cycle (*Briggs, 1974*). Aside from landmasses, long extensions of deep oceanic water and abrupt changes of physical or chemical properties of marine water can effectively restrict the colonization potential of inshore organisms (*Floeter et al., 2008*; *Luiz et al., 2011*). Furthermore, the effectiveness of these permeable barriers may be influenced by the potential to establish new populations in the recently colonized habitats (*Luiz et al., 2011*).

The mid Atlantic barrier (MAB) is a deep ocean gap formed by the Atlantic Ocean basin in the last 85 Myr. It spans a long distance with a minimum straight-line distance of approximately 2800 km, therefore acting as a barrier to larval dispersal for marine organisms (*Floeter et al., 2008*; *Luiz et al., 2011*).

For molluscs, *Vermeij (2005)* concluded that 30.8% (northeastern Atlantic) to 47.3% (northwestern Atlantic) of the North-Atlantic species present an amphi-Atlantic distribution. However, *F. picta* is the Atlanto-Mediterranean Chromodorididae with the wider distribution area (*Ortea, Valdés & García-Gómez, 1996*) and one of the few present in both margins of the Atlantic. the Mediterranean and most islands of the tropical and temperate north Atlantic, having crossed major biogeographic barriers. With such an extensive geographical distribution, several subspecies were historically assigned to *F. picta.* Some of these subspecies are morphologically very similar and present overlapping geographical distributions (e.g. *F. picta picta* and *F. picta webbi* cf. *Ortea, Valdés & García-Gómez, 1996*). A molecular phylogeny of this group would shed some light on the taxonomic status of *F. picta* subspecies and other closely related Atlanto-Mediterranean *Felimare* species.

In the present study, we intend to clarify the taxonomic status of *F. picta* and its relationships with other congeneric species. Simultaneously, we want to evaluate the validity of its previously proposed subspecies that currently raise numerous identification problems to many taxonomists. The molecular phylogeny will be the starting point to infer the biogeographic relationships within a group of taxa that was able to cross the main Atlantic biogeographic barriers.

## MATERIAL AND METHODS

### Sampling

The species sampled in the present study, the geographical origin of the samples and the GenBank accession numbers are listed in Table S2 in the Supplementary Data. Specimens were identified and portions of tissue were provided by a nudibranch taxonomist (Dr. Gonçalo Calado–see *Gaspar, Rodrigues & Calado (2009)*; *Coelho & Calado (2010)*; *Haber et al. (2010)*; *Calado & Silva (2012)*; *Cruz, Gaspar & Calado (2012)*). The same nudibranch taxonomist kindly provided pictures of several *Felimare* species and *F. picta* subspecies (Supplementary Data). Extracted DNA is available from ISPA laboratory collections. Voucher specimens of *F. lajensis* are available from Museu de Zoologia da Universidade de São Paulo (MZSP97468). In an attempt to detect possible intraspecific variability in this species, a total of 32 *F. picta* representing samples from its entire geographical distribution area and either side of the MAB were analysed. Additional sampling included the Strait of Gibraltar and the Mauritanian cold water barrier limiting dispersal in other marine species (e.g. the Strait of Gibraltar) (see *Patarnello, Volckaert & Castilho, 2007*). Samples belonging to 13 different *Felimare* species were also collected. These represent approximately half of the described species of the eastern Pacific, Atlantic and Mediterranean (*WoRMS Editorial Board, 2014*), including some already available in GenBank, which were used for comparative purposes to provide a broader phylogenetic framework and access the overall genetic divergence within this genus.

### DNA extraction, amplification and sequencing

DNA was extracted from tissue samples preserved in ethanol, using a proteinase K/SDS based protocol (*Sambrook, Fritsch & Maniatis, 1989*). Primers used to amplify a fragment 461 bp long from the 16S mitochondrial rDNA (16Ssar 5′ CGC CTG TTT ATC AAA AAC AT 3′ and 16Sbr 5′ CCG GTC TGA ACT CAG ATC ACG T 3′), and a fragment 554 bp long from the COI (LCO1490 5′ GGT CAA CAA ATC ATA AAG ATA TTG G 3′ and HCO2198 5′ TAA ACT TCA GGG TGA CCA AAA AAT CA 3′), are described in *Palumbi et al. (1996)* & *Folmer et al. (1994)*, respectively. The primers 28SC1(F) 5′ ACC CGC TGA ATT TAA GCA T 3′ and 28SD3(R) 5′ GAC GAT CGA TTT GCA CGT CA 3′ used by *Mollaret et al. (1997)*, *Vonnemann et al. (2005)* & *Klussmann-Kolb et al. (2008)* were also used to attempt to amplify a fragment of the nuclear 28S rDNA in these species.

PCR conditions were conducted as follows: 2 min. 95 °C followed by 35 cycles of (95 °C (30 sec.), 54 °C (30 sec.) and 72 °C (60 sec.)) for the 16S fragment; 2 min. 95 °C followed by 35 cycles of (95 °C (45 sec.), 50 °C (45 sec.) and 72 °C (60 sec.)) for the COI fragment and 4 min. 95 °C followed by 38 cycles of (94 °C (30 sec.), 52 °C (50 sec.) and 72 °C (120 sec.)) followed by 10 min. at 72 °C for the 28S fragment.

PCR products were purified using microClean (MicroZone, www.microzone.co.uk), and sequenced in STABVIDA (http://www.stabvida.net/) using these primers.

### Phylogenetic analysis

Following *Johnson (2011)* we used an outgroup species from the Dendrodorididae family: *Doriopsilla pelseneeri* d'Oliveira, 1895. DNA sequences were analysed and edited using the

software program CodonCode aligner (v. 3.5, CodonCode Corporation) and were aligned separately using M-Coffee (*Notredame, Higgins & Heringa, 2000*). Manual alignment masking was performed, by excluding loci scored as 'bad' by M-Coffee, in order to improve sign-to-noise ratio. Transitional saturation was examined by plotting transitions and transversions against sequence divergence using GTR distance and implementing Xia *et al.* test (*Xia et al., 2003*; *Xia & Lemey, 2009*) test of substitution saturation available in Dambe v. 5.3.108 (*Xia, 2013*).

Each fragment and a concatenation of both fragments were analysed using four phylogenetic inference methods: 1) maximum-parsimony (MP) with 100 heuristic searches using random additions of sequences and implementing the TBR algorithm, as implemented in PAUP 4.0b10 (*Swofford, 2001*); 2) minimum-evolution (ME), also implemented in PAUP with 1000 resamplings, was implemented using the best-fit model of molecular evolution chosen according the Bayesian Information Criterion as implemented in JModeltest 2.0 (*Darriba et al., 2012*); 3) Maximum Likelihood, as implemented in RaxML (*Stamatakis, Hoover & Rougemont, 2008*) and 4) Bayesian inference (BI) performed using MCMC as implemented in MrBayes v. 3.2 (*Ronquist et al., 2012*), with two independent runs of four Metropolis-coupled chains of four million generations each in order to estimate the posterior probability distribution. Topologies were sampled every 100 generations and a majority-rule consensus tree was estimated after discarding the first 10% samples. Convergence was verified by inspecting the average standard deviation of split frequencies and tracing likelihood throughout samples in Tracer v1.6 (*Drummond et al., 2012*). Both ML and BI analyses of the concatenated alignment considered two partitions for which independent parameters were estimated. For the first three phylogenetic inference methods, branch support values for each node were tested by bootstrap analysis, with 100 resamplings (*Felsenstein, 1985*). Net between group mean distances were calculated using Mega (*Tamura et al., 2013*) using Tamura-Nei distance with gamma model estimated by the composite likelihood method.

## RESULTS

### Sequence analysis

The null hypothesis of congruence between the two data sets (16S and COI rDNA) was not rejected ($P = 0.33$) by the ILD test (*Farris et al., 1995*). Therefore, the results presented in subsequent sections relate to the analysis of the concatenation of the 16S and COI rDNA fragments. The combined sequence of the 16S + COI rDNA fragments resulted in an alignment of 1026 base pairs. Of these, 712 characters are constant, 66 variable characters are parsimony-uninformative and 248 are parsimony-informative characters. No saturation was observed for the 16S and COI datasets or the concatenated fragment with both sequences ($P < 0.001$ for all combinations) (*Xia et al., 2003*; *Xia & Lemey, 2009*).

Minimum-evolution (ME), using the best-fit model of molecular evolution chosen according the Bayesian Information Criterion as implemented in JModeltest 2.0 (*Darriba et al., 2012*) was HKY + I + G for 16S; TrN + I + G for COI and TIM3 + I + G for the concatenated 16S and COI.

Since we were not able to amplify the COI fragment of *F. acriba*, known from the Caribbean, this species is not shown in the concatenated tree with the 16S and COI fragments (Fig. 1). However, all phylogenetic analysis with the 16S fragment recovered this species as the sister species of *F. bayeri* with very high support values (Bayesian analysis with posterior probability of 1.0 and maximum parsimony with bootstrap value of 100).

Estimated net evolutionary divergence between species and subspecies is presented in Table 1. Genetic similarities and haplotypes shared between *F. picta* collected from Mexico to the Mediterranean, including the Azores (*F. picta picta*, *F. picta webbi* and *F. picta azorica* sensu *Ortea, Valdés & García-Gómez, 1996*) revealed no genetic isolation. The genetic distance between these subspecies and the west African samples (0.255–0.272), which would be classified as *F. p. tema* and *F. p. verdensis* (sensu *Ortea, Valdés & García-Gómez, 1996*) is larger than the distance shown for *F. zebra* (0.185) and is similar to the one shown for *F. bayeri* (0.277).

## Phylogenetic analysis

### The genus Felimare and higher order relationships

The results presented in Fig. 1A support the distinctiveness of the eastern Pacific, Atlantic and Mediterranean *Felimare* species compared to the Indo-Pacific *Hypselodoris* species. The Caribbean *Felimare kempfi* failed to be included in a distinct clade with the remaining species of this genus.

### The Felimare picta complex

The monophyly of *F. picta*, including specimens from all geographical areas, is not supported given the internal position of several west Atlantic species, such as *F. zebra* and *F. lajensis* (Fig. 1B). Although ME and BI recovered a third west Atlantic taxa, *F. bayeri*, as the sister species of the African subspecies of *F. picta* (*F. p. tema* and *F. p. verdensis*), this phylogenetic relationship was not supported by ML or MP methods. As a precautionary measure these incongruent results were interpreted as an unresolved trichotomy (Fig. 1B). Nevertheless, *F. bayeri* was always recovered in a clade, including the remaining taxa of the *F. picta* complex, with very high support values. These results alone show that *F. picta*, as currently defined, constitutes a paraphyletic group.

The results presented in Fig. 1B also confirm that, unlike other species of the same genus, the distribution range of *F. picta* encompasses both margins of the Atlantic Ocean together with several North Atlantic islands, namely the Azores. Interestingly, samples from this central Atlantic archipelago had identical or similar haplotypes compared to others from Mexico (west Atlantic), Portugal and Spain (east Atlantic) or Italy (Mediterranean).

The paraphyly of *F. picta* and, consequently, the phylogenetic relationships between its subspecies and other related taxa requires a revision of the taxonomic status of this complex of species and raises the question of how many transatlantic colonisation events could have occurred within the *F. picta* complex alone.

For identification purposes, *F. picta* complex diagnostic nucleotide characteres are listed for 16S and COI fragments in Table S3 in the Supplementary Data.

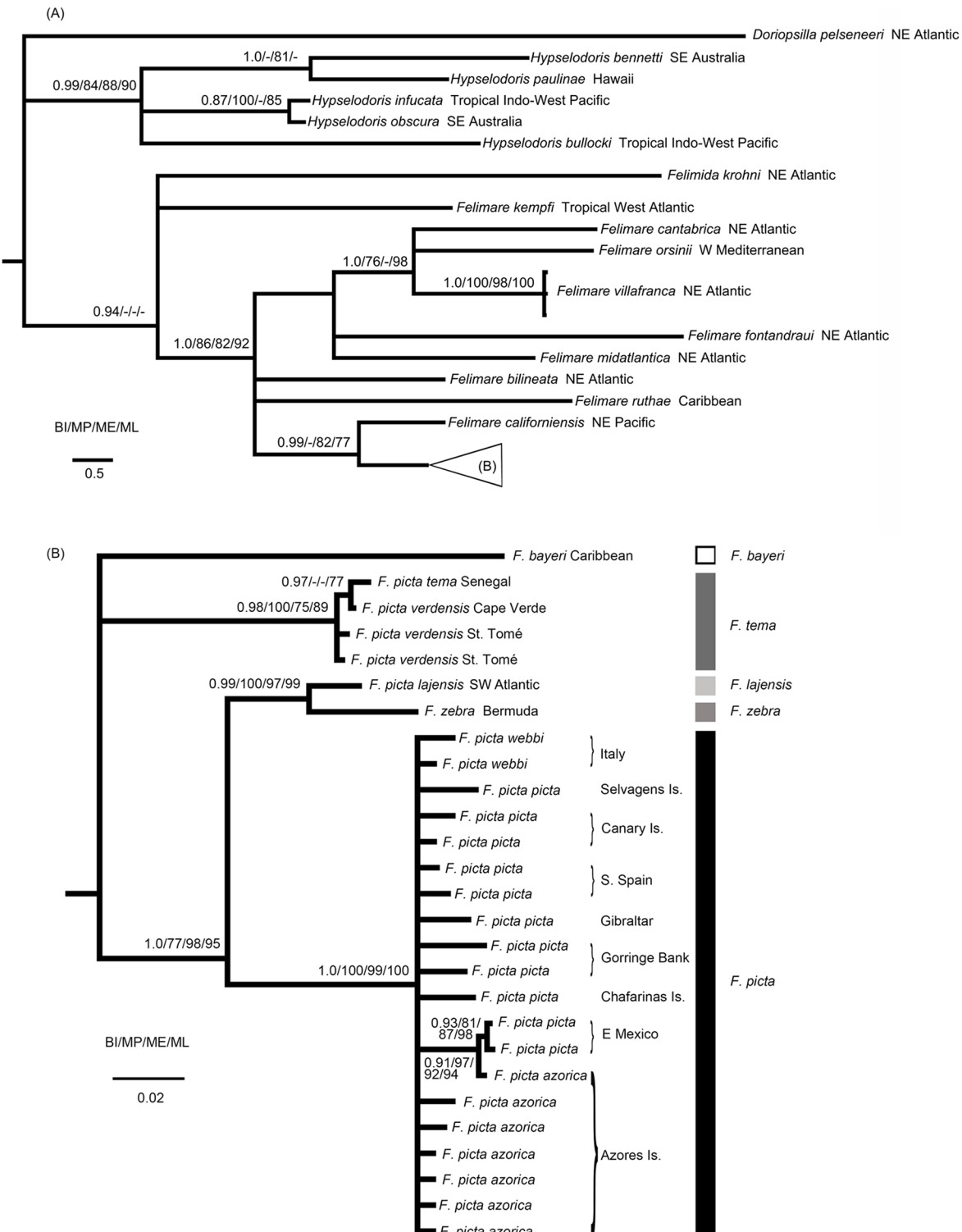

**Figure 1 Phylogenetic relationships of the Atlanto-Mediterranean "*F. picta* complex" and other related species.** (A) Phylogenetic relationships between several Indo-West Pacific *Hypselodoris* species and eastern Pacific, Atlantic and Mediterranean *Felimare* species obtained from the 16S + COI molecular data. Bayesian posterior probabilities (BI) and MP, ME and ML bootstrap support are shown above and below each node, respectively. (B) Phylogenetic relationships of the Atlanto-Mediterranean "*F. picta* complex."

**Table 1 Estimates of net evolutionary divergence.** Estimates of net evolutionary divergence between groups of sequences from *F. picta* subspecies, other Atlanto-Mediterranean *Felimare* species and Indo-West Pacific *Hypselodoris* species. Standard error estimate(s) are shown above the diagonal. The number of base substitutions per site from estimation of net average between groups of sequences are shown. The *Felimare picta* complex is highlighted in bold. Analyses were conducted using the Maximum Composite Likelihood model (*Tamura, Nei & Kumar, 2004*). The rate variation among sites was modeled with a gamma distribution (shape parameter = 0.426). The analysis involved 45 mtDNA sequences and all ambiguous positions were removed resulting in a total of 1124 nucleotides in the final dataset. Evolutionary analyses were conducted in MEGA6 (*Tamura et al., 2013*).

| | Chromodoris | F. bayeri | F. bilineata | F. californensis | F. cantabrica | F. fontadraui | F. kempfi | F. lajensis | F. midatlantica | F. orsinii | **F. picta azorica** | **F. picta picta** | **F. picta webbi** | **F. picta tema** | **F. picta verdensis** | F. ruthae | F. villafranca | F. zebra | Hypselodoris |
|---|---|---|---|---|---|---|---|---|---|---|---|---|---|---|---|---|---|---|---|
| Chromodoris | | 0,546 | 0,617 | 0,613 | 0,735 | 0,631 | 0,496 | 0,628 | 0,560 | 0,624 | 0,690 | 0,702 | 0,691 | 0,565 | 0,560 | 0,872 | 0,576 | 0,591 | 0,625 |
| F. bayeri | 1,766 | | 0,284 | 0,218 | 0,470 | 0,395 | 0,489 | 0,079 | 0,307 | 0,389 | 0,082 | 0,084 | 0,083 | 0,067 | 0,063 | 0,582 | 0,370 | 0,082 | 0,614 |
| F. bilineata | 2,055 | 0,922 | | 0,250 | 0,464 | 0,379 | 0,561 | 0,288 | 0,286 | 0,371 | 0,289 | 0,290 | 0,273 | 0,278 | 0,269 | 0,502 | 0,420 | 0,300 | 0,623 |
| F. californensis | 2,069 | 0,692 | 0,771 | | 0,442 | 0,363 | 0,648 | 0,199 | 0,330 | 0,382 | 0,197 | 0,197 | 0,192 | 0,182 | 0,179 | 0,630 | 0,363 | 0,209 | 0,769 |
| F. cantabrica | 2,592 | 1,536 | 1,520 | 1,454 | | 0,478 | 0,786 | 0,449 | 0,461 | 0,405 | 0,422 | 0,432 | 0,427 | 0,444 | 0,429 | 0,747 | 0,280 | 0,494 | 0,910 |
| F. fontadraui | 2,095 | 1,260 | 1,227 | 1,162 | 1,596 | | 0,633 | 0,453 | 0,309 | 0,329 | 0,481 | 0,479 | 0,480 | 0,395 | 0,381 | 0,602 | 0,399 | 0,441 | 0,700 |
| F. kempfi | 1,660 | 1,611 | 1,865 | 2,211 | 2,752 | 2,217 | | 0,519 | 0,586 | 0,653 | 0,576 | 0,577 | 0,572 | 0,533 | 0,522 | 0,845 | 0,679 | 0,542 | 0,702 |
| F. lajensis | 2,138 | 0,258 | 0,930 | 0,644 | 1,485 | 1,499 | 1,749 | | 0,338 | 0,458 | 0,050 | 0,050 | 0,050 | 0,074 | 0,074 | 0,652 | 0,346 | 0,022 | 0,749 |
| F. midatlantica | 1,839 | 0,941 | 0,919 | 1,053 | 1,497 | 0,968 | 1,973 | 1,085 | | 0,356 | 0,362 | 0,365 | 0,370 | 0,316 | 0,320 | 0,614 | 0,270 | 0,373 | 0,674 |
| F. orsinii | 2,044 | 1,194 | 1,175 | 1,206 | 1,373 | 1,059 | 2,208 | 1,489 | 1,147 | | 0,487 | 0,490 | 0,484 | 0,381 | 0,386 | 0,627 | 0,206 | 0,440 | 0,689 |
| **F. picta azorica** | 2,340 | 0,272 | 0,944 | 0,627 | 1,364 | 1,572 | 1,966 | 0,163 | 1,130 | 1,579 | | **0,000** | **0,000** | **0,081** | **0,078** | 0,651 | 0,408 | 0,054 | 0,734 |
| **F. picta picta** | 2,379 | 0,277 | 0,949 | 0,628 | 1,405 | 1,564 | 1,973 | 0,165 | 1,138 | 1,591 | **0,000** | | **0,000** | **0,083** | **0,080** | 0,647 | 0,412 | 0,053 | 0,732 |
| **F. picta webbi** | 2,338 | 0,272 | 0,891 | 0,617 | 1,387 | 1,569 | 1,957 | 0,164 | 1,155 | 1,569 | **0,000** | **0,000** | | **0,084** | **0,080** | 0,636 | 0,404 | 0,055 | 0,709 |
| **F. picta tema** | 1,847 | 0,211 | 0,885 | 0,572 | 1,450 | 1,259 | 1,798 | 0,246 | 0,994 | 1,180 | **0,262** | **0,269** | **0,272** | | **0,004** | 0,614 | 0,404 | 0,089 | 0,690 |
| **F. picta verdensis** | 1,826 | 0,200 | 0,866 | 0,570 | 1,406 | 1,207 | 1,767 | 0,250 | 1,019 | 1,218 | **0,255** | **0,259** | **0,260** | **0,008** | | 0,600 | 0,397 | 0,090 | 0,687 |
| F. ruthae | 3,040 | 1,936 | 1,691 | 2,161 | 2,538 | 2,027 | 2,945 | 2,218 | 2,051 | 2,156 | 2,212 | 2,206 | 2,161 | 2,090 | 2,035 | | 0,620 | 0,614 | 0,850 |
| F. villafranca | 1,920 | 1,128 | 1,313 | 1,124 | 0,901 | 1,274 | 2,339 | 1,068 | 0,830 | 0,635 | 1,269 | 1,283 | 1,256 | 1,263 | 1,247 | 2,112 | | 0,362 | 0,636 |
| F. zebra | 1,942 | 0,258 | 0,943 | 0,650 | 1,617 | 1,441 | 1,841 | 0,072 | 1,192 | 1,388 | 0,181 | 0,179 | 0,185 | 0,281 | 0,287 | 2,045 | 1,100 | | 0,701 |
| Hypselodoris | 2,209 | 2,135 | 2,113 | 2,736 | 3,266 | 2,459 | 2,473 | 2,693 | 2,350 | 2,401 | 2,585 | 2,577 | 2,484 | 2,423 | 2,424 | 2,978 | 2,182 | 2,463 | |

## DISCUSSION

Molecular phylogenetic relationships suggest that current taxonomy of the genus *Felimare* and, in particular, the *F. picta* subspecies should be revised.

Currently, *F. picta* forms a paraphyletic group since *F. lajensis*, formerly classsified as *F. picta lajensis*, is the sister species of *F. zebra*, another west Atlantic *Felimare* species. West African *F. tema*, formerly described as an *F. picta* subspecies, should be reinstated as an independent species. *F. picta*, the only chomodorid with a distribution area encompassing both margins of the Atlantic Ocean, including the Archipelago of the Azores, and the Mediterranean Sea, includes former *F. picta picta*, *F picta webbi* and *F. picta azorica*. No genetic structure was detected between these putative subspecies. The taxonomic and biogeographic implications of these proposals are addressed in the sections below.

### Taxonomic implications

Although presenting a distinct color pattern when compared to other *F. picta*, the genetic data presented here suggests that the Azorean samples are not isolated from the west or east Atlantic and Mediterraneam samples. Although differences in phenotype may result from recent genetic divergence or distinct ecological influences, the classification of taxa into different species or subspecies should be accompanied by genetic studies that are able to demonstrate their monophyly. The results are compatible with two alternative hypotheses: (i) genetic isolation between *F. picta* subspecies was recent, requiring fast evolving genetic markers, such as microsatellites, to detect reproductive isolation or; (ii) at least some of these "subspecies" coexist and interbreed. Considering the paraphyly of the *F. picta* complex and the fact that (*Ortea, Valdés & García-Gómez, 1996*) did not describe valid morphological characters or designate type specimens to justify the nomination of several new subspecies (e.g. *F. p. azorica*), *F. p. picta*, *F. p. webbi* or *F. p. azorica* should, by order of precedence and for the time being, be synonymized under *F. picta* (Schultz, 1836) instead of assigning them to different subspecies. This is particularly important within the chromodorids, which are known to be quite diversified in terms of color patterns. In fact, some of the characters described for some subspecies, such as the absence of yellow lines in *F. p. azorica*, are also observed in *F. p. picta* and the lack of discriminating differences is even more evident between the latter and *F. p. webbi*.

*F. p. tema* (*Edmunds, 1981*) and *F. p. verdensis* (*Ortea, Valdés & García-Gómez, 1996*) show no genetic differences between samples collected from Senegal and from Cape Verde. These results are not surprising since (*Ortea, Valdés & Garcia-Gómez, 1996*) reclassified *F. tema*, originally described by *Edmunds (1981)* from Ghana as a subspecies of *F. picta*, and described a new subspecies: *F. p. verdensis*, endemic to Cape Verde Islands, São Tomé and southern Angola. However, they did not designate any type specimens and the morphological variability described by *Edmunds (1981)* for *F. tema* is similar to that described for the specimens of *F. p. tema* and *F. p. verdensis* except for one difference related to radula secondary denticles which are absent in *F. p. tema*. These specimens are characterized by longitudinal orange lines, orange submarginal border, orange lined gills and a dark blue background color with lighter blue patches along the submarginal mantle border.

The results presented here suggest that, for the time being, the original taxonomic classification by *Edmunds (1981)* should be reinstated and *F. tema* should include both *F. p. tema* and *F. p. verdensis*. In fact, this tropical eastern Atlantic species, whose distribution area would now include Senegal, Ghana, south Angola and the archipelagoes of Cape Verde and São Tomé, is more distantly related to other parapatric norheastern Atlantic *F. picta* than the allopatric western Atlantic *F. lajensis*.

*F. p. lajensis Troncoso, Garcia & Urgorri, (1998)* was originally described for Brazil as a subspecies of *F. picta* but has been re-assessed as a different species by *Domínguez, García & Troncoso (2006)*. Its ambiguous taxonomic status (species or subspecies) is rooted on the morphological descriptions that: i) distinguish this taxon from remaining *F. picta*, based on the dark blue to violet body pattern with blue to violet gills, five spaced dorsal lines and a deferent duct with a narrow preprostatic portion (*Troncoso, Garcia & Urgorri, 1998*; *Domínguez, Garcia & Troncoso, 2006*) or ii) suggest that it should be included in *F. picta* because some specimens have gill rachis with yellow lines, up to nine dorsal yellow lines and a yellow or white mantle margin (*DaCosta, Padula & Schrödl, 2010*). In fact, *F. picta* present gill rachis with yellow lines, only three dorsal lines and the preprostatic portion of the deferent duct is absent. *DaCosta, Padula & Schrödl (2010)* also showed that radular and allosperm receptacles differences described by *Troncoso, Garcia & Urgorri (1998)* & *Domínguez, García & Troncoso (2006)* were not supported by the additional material analysed by those authors. Due to the wide morphological variation *DaCosta, Padula & Schrödl (2010)* decided to keep this taxon as a subspecies of *F. picta* until detailed anatomical comparisons and also molecular approaches are concluded to better understand not only the satus of this taxon but also the relationships between all *F. picta* subspecies.

In our study, *F. picta* samples from Brazil were always recovered in a clade with *F. zebra* (Bermuda) and the genetic distance suggests a close relationship between these taxa. This southwestern Atlantic taxon is the sister clade to all remaining *F. picta*, excluding the east tropical Atlantic *F. tema* which diverged near the base of the *F. picta* complex as did also *F. bayeri*.

Although our results using two mitochondrial markers are clear and unambiguous, phylogenies based on mitochondrial DNA may result in misleading speciation histories when there are discrepancies with nuclear DNA (e.g. *Zhang & Hewitt, 1996*; *Shaw, 2002*). Joint analysis of mitochondrial and nuclear DNA may be informative even when results from both types of fragments are contradictory, particularly "when the context for the conflict is understood" (*Rubinoff & Holland, 2005*). In fact, nuclear phylogenies frequently reinforce mitochondrial phylogenies (e.g. *Levy et al., 2011*). On the other hand, nuclear loci present several technical limitations in phylogenetic inference such as low copy number, heterozygosity, paralogous loci with multiple copies and low substitution rate, which make them uninformative when comparing close related species or even subspecies (for a revision see *Rubinoff & Holland, 2005*). One example of this argument are the chromodorid species analysed by *Ortigosa et al. (2014)* where nuclear markers did not add any additional information or resolve ambiguous results on closely related *Felimida* species. Although, reclassifying *F. picta*, *F. tema* and *F. lajensis* as independent species

should be considered provisional until more specimens are throughly analysed and nuclear markers are compared, the suggestions presented above clarify the taxonomy and help to define groups that are monophyletic and well characterized with mitochondrial DNA markers and morphological characters.

In a wider phylogenetic scope the results indicate a basal politomy with four branches. Western Atlantic *F. ruthae* and eastern Atlantic/west Mediterranean *F. bilineata* appear as individualized branches. A third clade includes the *F. picta* complex, with the amphi-Atlantic *F. picta*, western Atlantic *F. lajensis*, *F. zebra* and *F. bayeri* and eastern Atlantic *F. tema* and its sister species, the eastern Pacific *F. californiensis*. Future phylogenetic studies will show if other eastern Pacific species are included in the same or in different clades together with the remaining Atlantic and Mediterranean species. A forth clade includes the remaining eastern Atlantic and Mediterranean species: *F. cantabrica*, *F. fontandraui*, *F. orsinii*, *F. midatlantica* and *F. villafranca*. The exclusion of *F. kempfi* from the *Felimare* clade was already argued by *Johnson & Gosliner (2012)* placing this Caribbean species with the eastern Pacific *F. porterae* as a potential sister group to a larger clade of eastern Pacific, Atlantic and Mediterranean *Felimare* species (*Johnson & Gosliner, 2012*).

The fact that the Indo-Pacific species are included in a distinct monophyletic clade corroborates the conclusions of *Johnson & Gosliner (2012)* pointing to an ancient diversification of the *Felimare* species in the eastern Pacific, Atlantic and Mediterranean since the closure of the Tethys Sea in the East by the end of the Miocene.

## Felimare picta complex biogeography

Sea surface currents and larvae characteristics, including larval behaviour, are crucial for an organism to be able to cross important biogeographic barriers. *Briggs (1995)* described four biogeographic provinces later reviewed by *Spalding et al. (2007)* & *Briggs & Bowen (2012)* that are relevant to this study: northeastern Atlantic Lusitania (including Macaronesia and the Mediterranean) (but see *Almada et al., 2013*), tropical West Africa, the Caribbean and Brazil. These provinces are delimited by several soft (non-terrestrial) barriers such as: (1) the mid-Atlantic barrier; (2) the Mauritanian cold water upwelling in the northwestern coast of Africa; (3) the Orinoco/Amazonas freshwater plume; (4) the Almeria/Oran front separating the Atlantic from the Mediterranean. Some organisms can cross these areas when environmental conditions are favourable which result in the establishment of new populations or species (e.g. *Floeter et al., 2008*).

Based on the phylogenetic results presented in this study the success of the *F. picta* complex crossing all these biogeographic barriers underlined by the amphi-Atlantic distribution of *F. picta* is not reflected among the other chomodorids.

## Panama isthmus

The high diversity in the eastern Pacific and the opportunity to colonize the western Atlantic via Panama Isthmus before its closing, contributed to a higher species richness in this area compared to other Atlantic regions. This fact allowed not only for species to exist on both sides of the isthmus, but also some taxa to radiate to many others (*Lessios, 2008*).

Examples of sister relationships across this particular vicariant event include opisthobranchs (e.g. *Malaquias & Reid, 2009*) and fish (*Briggs, 1995*; *Muss et al., 2001*; *Grant, Lecomte & Bowen, 2010*). This pattern is also observed between the eastern Pacific *F. californiensis* and the *F. picta* complex. In fact, the *F. picta* complex is more closely related with *F. californiensis* than with other Atlanto-Mediterranean *Felimare* species which probably means that the origin of this species complex is posterior to the closure of this hard (terrestrial) biogeographic barrier.

## Mid-Atlantic barrier

The Atlantic ocean barrier is an important constraint to the migration of individuals between both sides of the Atlantic (*Briggs, 1995*). However, crossing of the MAB by some species has been suggested by the amphiatlantic distribution of these species or because they have sister taxa on both margins of the Atlantic reflecting historical speciation events (*Carmona et al., 2011*).

The genus *Felimare* and, in particular, the *F. picta* complex, illustrates both these patterns. Present geographic distribution of *F. picta* includes the west and eastern Atlantic Ocean and also the Mediterranean Sea which is supported by molecular data presented here. Although this hypothesis should be tested in the future with additional samples and appropriate phylogeographic analysis, the phylogeny of this group suggests that the mid-Atlantic barrier was probably crossed twice by this complex of species. First, an earlier colonization event resulted in the divergence of *F. tema* (including *F. picta tema* and *F. picta verdensis*) in equatorial Africa region. Second, a separation between the clade *F. lajensis/F. zebra* and *F. picta* (including *F. picta* from western and northeastern Atlantic) and a transatlantic migration resulting in the colonization of both sides of the Atlantic by this later species.

Major oceanic surface currents suggest that this migration could have followed a westward route, particularly in the equatorial region (e.g. *Silva, Horne & Castilho, 2014* and references therein). With predominant surface currents from northwest Africa to Central America (North equatorial current) and from southwest Africa to southeast South America (South equatorial current), one hypothesis to explain current species distribution would be an westward migration with speciation along the American coast. However this westward migration hypothesis is highly improbable if we consider the phylogeny of this group of species, particularly the basal position of the west Atlantic species within the *F. picta* complex phylogeny and the eastern Pacific *F. californiensis* being the sister species of this clade.

An alternative hypothesis would be an eastward colonization from the western Atlantic to the European and African coasts which, if we assume that the present surface current pattern was already in effect, could follow two alternative routes: a northern route along the Gulf stream followed by the Azores and the Canaries currents (*Barton, 2001*) and an equatorial route following the north equatorial countercurrent (see *Fonseca et al., 2004*). Phylogenetic patterns showed in this work and those reported by other authors based on morphological and meristic data (*Gosliner & Johnson, 1999*; *Alejandrino & Valdés, 2006*) also support this eastward migration hypothesis.

Assuming that the origin of *F. picta* is posterior to the closure of the Isthmus of Panama and therefore posterior to the settlement of the Gulf Stream, the similarity between Mexican samples of *F. picta* and an individual collected in the Azores suggest that this archipelago may have acted, and still acts, as a stepping-stone in this northern route. This hypothesis is further supported by the fact that during glacial periods these currents were even stronger than in present times (*Wunsch, 2003*) which could result in the rapid transportation from west to the east Atlantic of planktotrophic larvae or adults and eggs on rafting materials. The equatorial counter current is seasonal, being stronger during the Spring (*Richardson et al., 1992*), when it reaches surface transport velocities between 23 cm s$^{-1}$ (*Richardson et al., 1992*) and 45 cm s$^{-1}$ (*Urbano, Almeida & Nobre, 2008*). This would mean that the mid Atlantic barrier could be crossed at maximum velocity in less than 10 days (38.88 km day$^{-1}$). This would allow an independent eastward migration by a southern route which could have led to the origin of *F. tema* in the tropical west African coast. Only future phylogeographic studies with population samples of *F. picta* from both sides of the Atlantic may shed some light on the dispersion route of this species.

## Mauritanian cold water barrier

The present allopatric distribution of *F. tema* and *F. picta* could be explained by the persistence of biogeographic barriers and/or by ecological constraints. The cold water barrier along the Mauritanian shores due to strong upwelling (*Marañón et al., 2001*) and the Pleistocenic glaciations could have prevented *F. tema* from colonizing the northeastern Atlantic shores. More recently, with the settling of *F. picta* in the northeastern Atlantic shores other ecological constraints may have been in effect. This argument is based on the fact that *F. tema* is deeply rooted within a clade with several west Atlantic extant species and *F. picta* shares a common ancestor with *F. lajensis* and *F. zebra*, which are also west Atlantic species. If the hypothesis of the "eastward migration" proves to be correct, present allopatric distribution of *F. tema* and *F. picta* may be the result of two independent dispersion processes: the first resulting in the speciation of the tropical east Atlantic *F. tema* from an west Atlantic ancestor, and the second resulting in the colonization of the northeastern Atlantic and the Mediterranean by *F. picta*.

Examples of this eastward migration are common across a large array of taxonomic groups and are much more common than migrations on the opposite directions (*Ávila, 2005*; *Rocha et al., 2008*; *Beldade et al., 2009*). This may be the result of the predominant current patterns described above and the higher species richness in the west Atlantic shores.

## Orinoco/Amazonas barrier

Genetic isolation between central western and southwestern Atlantic may have occurred about 6 million years with the origin of the Amazon River (*Nunan, 1992*; *Hoorn, 1994*). Variation in salinity levels due to river fresh water plumes and the absence of appropriate hard substrate along the southwest Atlantic coast became known as the Central American Gap and represent an important biogeographic barrier for marine organisms (*Rocha, 2003*;

*Ludt & Rocha, 2014*). Since the closure of the Isthmus of Panamá the patterns of epipelagic circulation remained approximately constant (*Haug & Tiedemann, 1998*) therefore, conditions were appropriate for this biogeographic semi-permeable barrier to promote a recent split between *F. lajensis* and *F. zebra*.

The question still remains however on why among all chromodorids has the *F. picta* complex experienced such a success crossing all main Atlantic biogeographic barriers? Even considering *F. picta sensu strictu*, this is the only chromodorid whose distribution encompasses northwestern Atlantic, northeastern Atlantic and the Mediterranean. Futhermore, nudibranchs and the chromodorids, in particular, are sedentary and have a dispersion ability that is much reduced during all life stages. Rafting on floating materials could explain their ability to colonize distant locations; however, the individuals of these species are benthic and are usually found over sponges upon which they feed. Therefore, it is improbable that these organisms may disperse during their adult phase. *Coelho & Calado (2010)* reported that *F. picta* shows the largest egg size and planktotrophic larvae length at hatching reported among nudibranch molluscs (for a review see *Todd, Lambert & Davies, 2001*). Although *F. villafranca* have even larger eggs, it presents direct and not planktotrophic development. It is commonly accepted that large larvae have higher survival rates during transport in the water column. Although, *Shulman & Bermingham (1995)* found no relationship between oceanographic patterns, larval duration and population genetic structuring in the Caribbean, they refer that a different scenario could emerge on larger geographic scales. For this purpose, additional studies on different species which were also able to cross the MAB, such as the Chromodorididae *Tyrinna evelinae* and *Cadlina rumia*, from a sister group of the Chromodorididae (*Johnson, 2011*), could be of extreme interest as well as tagging mark and recapture of these species to determine if the distance they travel enables physical crossing.

From a molecular ecology perspective, genetic assignment tests could indicate if the most likely sources of recruits are from local or distant populations (see *Piry et al., 2004*; *Wilson & Rannala, 2003*).

A comprehensive study of the phylogeny of the genus *Felimare*, including all its species and nuclear DNA markers is still needed to clarify the taxonomy of this group. Furthermore, phylogeographic data would provide information on the direction and number of colonization events of each taxonomic entity and would allow the implementation of species delimitation analysis (see *Puillandre et al., 2012*).

Nevertheless the biogeographic considerations and phylogenetic relationships described above may help to refine current information on a group of marine organisms that have been raising the attention of a broad community from evolutionary biology and ecology to natural products chemistry.

## ACKNOWLEDGEMENTS

We would like to thank the help of Dr. Gonçalo Calado for all the expertise identifying nudibranchs specimens, for the pictures of several specimens and valuable comments on the manuscript. We would like to thank Rita Coelho and the Portuguese Institute of

Malacology for the pictures of several specimens. We would also like to thank the Museu de Zoologia da Universidade de São Paulo for samples under the material transfer agreement No. 003/2007/MZUSP with the Portuguese Institute of Malacology.

### Funding
This study had the support of the Fundação para a Ciência e a Tecnologia (FCT) (UID/MAR/04292/2013 and FA research SFRH/BPD/63170/2009). The funders had no role in study design, data collection and analysis, decision to publish, or preparation of the manuscript.

### Grant Disclosures
The following grant information was disclosed by the authors:
Fundação para a Ciência e a Tecnologia (FCT): UID/MAR/04292/2013.
FA research: SFRH/BPD/63170/2009.

### Competing Interests
The authors declare they have no competing interests.

### Author Contributions
- Frederico Almada conceived and designed the experiments, performed the experiments, analyzed the data, contributed reagents/materials/analysis tools, wrote the paper, prepared figures and/or tables, reviewed drafts of the paper.
- André Levy conceived and designed the experiments, performed the experiments, analyzed the data, contributed reagents/materials/analysis tools, wrote the paper, prepared figures and/or tables, reviewed drafts of the paper.
- Joana I. Robalo conceived and designed the experiments, analyzed the data, contributed reagents/materials/analysis tools, wrote the paper, reviewed drafts of the paper.

### DNA Deposition
The following information was supplied regarding the deposition of DNA sequences:
The GenBank accession numbers for the 16S and COI DNA sequences are available in Supplemental Table 2.

### Data Deposition
GenBank accession numbers:
16S – KT820489–KT820536
COI – KT833228–KT833269

### Supplemental Information
Supplemental information for this article can be found online at http://dx.doi.org/10.7717/peerj.1561#supplemental-information.

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
