# Peer review of "Not so sluggish: the success of the Felimare picta complex (Gastropoda, Nudibranchia) crossing Atlantic biogeographic barriers"

_PeerJ, doi:10.7717/peerj.1561_

## Round 0.1 · original submission · Major Revisions

Please consider all the suggestions of the reviewers in order to improve your manuscript.

Reviewer 1 ·

Basic reporting

The subject of the present manuscript is very interesting. However, some points I consider very important are missing: 1) deposit of examined specimens/vouchers in scientific collections; 2) color photos of the different species/subspecies; 3) species delimitation analyses in addition to the phylogenetic ones.
There are also some mistakes along the text which I highlighted directly in the pdf.

Experimental design

Absence of vouchers specimens/numbers
Absence of photos
Absence of species delimitation analyses (e.g. ABGD, GMYC...)

Validity of the findings

Section '4.2. Felimare picta complex biogeography' makes no sense if the authors think their data are not enough for a new taxonomic hypothesis on the F. picta complex.

Additional comments

The subject is very interesting but the study lacks some relevant points, mainly 1) deposit of examined specimens/vouchers in scientific collections; 2) color photos of the different species/subspecies; 3) species delimitation analyses in addition to the phylogenetic ones. There are also some mistakes along the text which where commented directly in the pdf.

Annotated reviews are not available for download in order to protect the identity of reviewers who chose to remain anonymous.

Reviewer 2 ·

Basic reporting

This is a pertinent study on the phylogeny and biogeography of the nudibranchia Felimare picta complex. This manuscript is of particular relevance to readers with interest on the study of biogeography and evolution of low dispersal marine invertebrates. The manuscript needs major changes, concerning the structure (e.g. line 244), English (e.g. line 281), objectivity (e.g. line 94: "Table 2 shows a list with some of these species."; some is not objective) and details (e.g. Table 1 - Brazil instead of Brasil). The inclusion of nuclear data would be of extreme importance to uncover the phylogeny and biogeography of the Felimare picta complex, but the authors did not ignore this fact and tried unsuccessfully to amplify the 28S nuclear region. Marine molluscs nuclear DNA is difficult to amplify (Reece KS, Ribeiro WL, Gaffney PM, Carnegie RB, Allen SK. Microsatellite marker development and analysis in the eastern oyster (Crassostrea virginica): confirmation of null alleles and non-mendelian segregation ratios. J Hered. 2004;95:346–352) and therefore the decision to include nuclear markers should be taken by the editor. I strongly encourage the authors to improve their manuscript taking into consideration the suggestions of the reviewers and re-submit it. I encourage the authors to address the following points:

Abstract:
The Abstract has contradictory information (e.g. long vs. short planktonic phase).

Introduction:
The Introduction should be revised. One of the goals stated by the authors is to reveal the biogeography of the F. picta complex. However, no reference to biogeographic patterns found in other marine organisms with low dispersal ability was mentioned. Phrases started in lines 39 and 45 are irrelevant to the study. The authors did not follow uniform criteria for geographic regions (e.g. west vs. western). Tables 1 and 2 may be presented as Supplementary Data and converted to a map. The objectives are clear, but no hypothesis or expectations are formulated conducting to a purely descriptive work. I suggest authors to hypothesize different biogeographic scenarios with possible routes of dispersal and timings of colonization.

Materials and Methods:
Table 3 should be presented in Supplementary Data and converted into a map with distribution ranges and sampling locations in the main document. The concatenation of mitochondrial sequences allowed increasing the length of the alignment and provided a stronger statistical support. The lack of nuclear markers is critical, but since the authors were not able to amplify the 28S fragment, the present data is a valuable contribution should be shared with the scientific community. Methods are congruent with the proposed objectives, however phylogenetic inference methods are too many (Only ML and BI should be presented) and lacks methods to support biogeographic events (e.g. Migrate). Also, a method to date lineage splitting-events should be used (e.g. BEAST).

Results
Some phrases should be transferred to the Discussion section (lines 244 and 253).

Discussion
The text needs structure: a brief mention of main results, caveats (e.g. lack of nuclear markers), discussion of hypothesis and expectations, possible explanations, support/refutation. The redrafting of the entire discussion according to well-stated objectives is critical.
The authors extensively described morphology without performing morphological analyses.

Experimental design

No hypothesis or expectations are formulated conducting to a purely descriptive work. Methods are congruent with the proposed objectives, however phylogenetic inference methods are too many (Only ML and BI should be presented) and lacks methods to support biogeographic events (e.g. Migrate). Also, a method to date lineage splitting-events should be used (e.g. BEAST).

Validity of the findings

Conclusions need to be supported on results and lacks important references.

Additional comments

Please find comments on the PDF file attached.

Annotated reviews are not available for download in order to protect the identity of reviewers who chose to remain anonymous.

Reviewer 3 ·

Basic reporting

Abstract
Line 6: Remove italics for genus Felimare. Italics should only be used if you are giving both the genus and species name unless you use for example, Felimare sp. Do this throughout the document.
Line 12-16: A long sentence, could it be separated into two?
Line 18: You mention that they have a short planktonic larval phase but then you counteract this in the final line 22 by saying that it has a long planktotrophic phase. Planktonic is dispersing in the plankton while planktotrophic means that it is plankton feeding and hence would also disperse via the plankton. Do you mean long in both lines as planktonic larvae are presumed to have longer larval durations as compared to direct developers?
Introduction
Line 45: What kinds of questions in regards to the phylogeny of chromodorids are obstacles for future investigation? Add further detail.
Line 51-53: You combine references for morphological and molecular studies (refs.), but do these citations include both a morphological and molecular component or are they refs. that are either morphological or molecular? If the latter, they must be separated into either refs. for molecular and separated into morphological refs. as well.
Line 57: Why was it identified as a potential barrier, to what? Your title is in relation to a biogeographic barrier and though you make reference to authors identifying it as a barrier, a little detail of the importance of the various barriers that exist is needed. I see you address this in the Discussion in lines 361 onwards until the end of this section but some of that description needs to be brought in sooner in the Introduction, Methods and even perhaps when describing the Results to make the manuscript more coherent.
Line 71: Left bracket ‘(‘ for Edmunds, 1981).
Line 98: Reference not in reference list for Valdés et al. 2006
Line 99: Table: Brackets for Marcus & Hughes, 1974. In the table you show the distribution areas for each Felimare species. I would include what barriers these are found across or what side of the barrier these belong to. You mention east and west Altantic etc. but also say that these are for example, east and west of the Altlantic barrier.
Line 101-117: This section needs additional detail and a more thorough description of what the mid Atlantic barrier is. What biogeographical, biotic and abiotic environmental factors (ocean current, temperature changes, lack of suitable habitat or land between the two sides of the barrier) attribute to it being recognised as an important barrier for example, to dispersal? Do not assume that the reader is familiar with your study region. What are the major or main barriers in the area and why. Would a map showing the barrier be relevant?
Line 101: Add references for …”highly mobile species (refs)”.
Line 116: Typo for ‘accross’ should be ‘across’.
Methods
Line 123: Are 32 samples enough to draw conclusions on pattern inferences in respect to barriers and colonization?
Line 124: How were you able to identify between all 13 species of Felimare. Do they have morphological traits that differed, or was it their distribution areas because they do overlap for one of the species (F. picta). Did life history phase (adult vs juvenile) make a difference to species identification?
Line 145: Space between 10min (10 min).
Line 240-244: Ok that you add the regions in brackets but you need to relate it back to the theme of barriers as you do state in your aims that you wanted to “infer the patterns of colonization of different regions accross the main Atlantic biogeographic barriers.”
Line 290: Should the two references have brackets around the year of publication or author and year?
Line 330: Comma after ‘In fact’.
Discussion
Line 328: Add an additional reference aside from (Shaw, 2002).
Line 452: Perhaps describe further what future experimental designs are necessary to tease apart the successful crossing of F. picta across all these barriers for example, do a recruit vs adult population genetic study and assignment tests. Also, is there additional literature for other kinds of invertebrates/molluscs that show a similar pattern of successful crossing over these barriers?
Please comment on any areas where the article fails to meet our standards, and in those cases please suggest ways in which it should be altered in order to meet the appropriate standard. All comments in this box will be sent to the author. If you have no comments on this aspect, just write "No Comments".
• The submission must adhere to all PeerJ policies.
Pass
• The article must be written in English using clear and unambiguous text and must conform to professional standards of courtesy and expression.
The article could use more scientific expression and less colloquial language but is fine as is and meets standards of courtesy and expression. Minor typo errors were present and references cited not added to the reference list (refer to ‘Basic reporting’) box for modifications required.
• The article should include sufficient introduction and background to demonstrate how the work fits into the broader field of knowledge. Relevant prior literature should be appropriately referenced.
Okay

• The structure of the submitted article should conform to one of the templates. Significant departures in structure should be made only if they significantly improve clarity or conform to a discipline-specific custom.
Pass
• Figures should be relevant to the content of the article, of sufficient resolution, and appropriately described and labeled.
Pass although a map highlighting the main biogeographic barriers of the area in relation to the sample sites would be beneficial to the reader for extra clarity.
• The submission should be ‘self-contained,’ should represent an appropriate ‘unit of publication’, and should include all results relevant to the hypothesis. Coherent bodies of work should not be inappropriately subdivided merely to increase publication count.
An additional paragraph in the results section linking the aim of “infer the patterns of colonization of different regions accross the main Atlantic biogeographic barriers.” (second aim) would be worthwhile. There are thorough results sections in the manuscript for the first aim (“we intend to clarify the taxonomic position of F. picta and its relationships with other congeneric species and simultaneously evaluate the validity of its recognised subspecies”) but lacking content to address the second aim. Perhaps further reference in the results section to the regions of interest that you identify as barriers will allow readers to draw more of a connection to the second aim.
• All appropriate raw data has been made available in accordance with our Data Sharing policy.

Experimental design

Refer to basic reporting for methods comments.

Validity of the findings

• The data should be robust, statistically sound, and controlled.

Pass

• The data on which the conclusions are based must be provided or made available in an acceptable discipline-specific repository.

Pass

• The conclusions should be appropriately stated, should be connected to the original question investigated, and should be limited to those supported by the results.

I am struggling to link the second aim to the appropriate results section. You address the biogeography well in discussion but this must be emphasized further in the results and throughout the manuscript otherwise, the paper becomes a study on teasing apart these genus and species discrepancies for the sea slugs only and not a study on barriers and the biogeography of the species.

Additional comments

It would be fantastic to do future studies examining the contemporary patterns of connectivity and running assignment tests for this species and look at migrant ancestry on programs like STRUCTURE.

---

## Round 0.2 · Minor Revisions

All 3 re-reviewers had comments regarding further improvements. Please consider all the suggestions in the revised manuscript.

Reviewer 1 ·

Basic reporting

The manuscript is now improved with changes and corrections but my previous three main highlights (1. absence of vouchers specimens/numbers; 2. absence of species photos; 3. absence of species delimitation analyses) were not improved by the authors. At least points 1 and 2 are crucial, in my point of view, for a relevant taxonomic manuscript and their absence weakens the manuscript. This is my opinion, but I will leave in the hands of the editor the decision to accept or not the current manuscript without the inclusion of such sections.
English text still needs revision, the use of "several" is not adequate in some situations.

Experimental design

1. absence of vouchers specimens/numbers; 2. absence of species photos

Validity of the findings

1. absence of vouchers specimens/numbers; 2. absence of species photos

Additional comments

The manuscript is now improved with changes and corrections but my previous three main highlights (1. absence of vouchers specimens/numbers; 2. absence of species photos; 3. absence of species delimitation analyses) were not improved by the authors. At least points 1 and 2 are crucial, in my point of view, for a relevant taxonomic manuscript and their absence weakens the manuscript. Concerning point 3, authors could look, at least, for diagnostic nucleotide characteres between F. picta, F. lajensis and F. tema.
English text still needs revision, e.g. the use of "several" is not adequate in some sentences.

Reviewer 2 ·

Basic reporting

This is a re-evaluation of a previous manuscript that was recommended to Major revisions. Authors mostly attended to reviewers recommendations and therefore I strongly recommend its publication after minor revisions. However, some parts of the text needs to be revised (e.g. line 24), others should be excluded or reduced (line 45), there are a lot of incongruences concerning abbreviations (e.g. F. picta vs. Felimare picta), italics, etc... From your words I did not understand if you considered the mid-Atlantic ridge as a barrier to dispersal as in Floeter et al., 2008, or the open-ocean with no suitable habitat for nudibranchs. I think that the manuscript could be significantly improved after an English native speaker correction. Also, I insist that the manuscript would be much easier to interpret if a map with sampling sites and major oceanic currents were presented.

Experimental design

Colonization events still lack a timeframe that could increase the interest of the manuscript and capture more audience. Although it is difficult to find nudibranch fossils, you could calibrate your tree with the closure of Panama as suggested by Grant et al. 2010.

Validity of the findings

No Comments.

Additional comments

Although I insist on the presentation of a map and on lineage-split events analyses, I let it to the editors consideration. Please find attached my suggestions.

Annotated reviews are not available for download in order to protect the identity of reviewers who chose to remain anonymous.

Reviewer 3 ·

Basic reporting

Dear Peer J and manuscript authors,
Improvements have been made to enhance previously concerning issues in this manuscript. My main concern now is the language used which needs refinement and careful wording. I have attached my comments and edit to the word document for author consideration to amend where necessary.

Experimental design

No comments

Validity of the findings

No comments

Additional comments

Please check wording carefully.

Annotated reviews are not available for download in order to protect the identity of reviewers who chose to remain anonymous.

---

## Round 0.3 · accepted · Accept

The authors have improved the manuscript and it is now Acceptable.